# Lactones in the Synthesis of Prostaglandins and Prostaglandin Analogs

**DOI:** 10.3390/ijms22041572

**Published:** 2021-02-04

**Authors:** Constantin Tănase, Lucia Pintilie, Raluca Elena Tănase

**Affiliations:** 1Synthesis of Biologically Active Substances Department, National Institute for Chemical-Pharmaceutical Research and Development, 74373 Bucharest, Romania; lucia.pintilie@gmail.com; 2Department of Mathematics, Angstrom Laboratory, Uppsala University, 751 06 Uppsala, Sweden; raluca.tanase@math.uu.se; 3“Simion Stoilow” Institute of Mathematics of the Romanian Academy, 010702 Bucharest, Romania

**Keywords:** 1,9-, 1,11-, and 1,15-PG lactones, δ-lactones, γ-lactones, prostaglandin intermediates, lactone synthesis, enzyme resolution, chemical resolution

## Abstract

In the total stereo-controlled synthesis of natural prostaglandins (PGs) and their structural analogs, a vast class of compounds and drugs, known as the lactones, are encountered in a few key steps to build the final molecule, as: δ-lactones, γ-lactones, and 1,9-, 1,11-, and 1,15-macrolactones. After the synthesis of 1,9-PGF_2α_ and 1,15-PGF_2α_ lactones, many 1,15-lactones of E_2_, E_3_, F_2_, F_3_, A_2_, and A_3_ were found in the marine mollusc *Tethys fimbria* and the quest for understanding their biological role stimulated the research on their synthesis. Then 1,9-, 1,11-, and 1,15-PG lactones of the drugs were synthesized as an alternative to the corresponding esters, and the first part of the paper describes the methods used for their synthesis. The efficient Corey procedure for the synthesis of prostaglandins uses the key δ-lactone and γ-lactone intermediates with three or four stereocenters on the cyclopentane fragment to link the PG side chains. The paper describes the most used procedures for the synthesis of the milestone δ-Corey-lactones and γ-Corey-lactones, their improvements, and some new promising methods, such as interesting, new stereo-controlled and catalyzed enantioselective reactions, and methods based on the chemical/enzymatic resolution of the compounds in different steps of the sequences. The many uses of δ-lactones not only for the synthesis of γ-lactones, but also for obtaining 9β-halogen-PGs and halogen-substituted cyclopentane intermediates, as synthons for new 9β-PG analogs and future applications, are also discussed.

## 1. Introduction

Lactones are widely encountered in nature (in plants and animals) as cyclic esters of hydroxy-acids in various forms from γ- [1], δ- [2], to larger rings of macrocyclic esters [3,4,5]. γ-lactones are estimated to be present in nearly “10% of all-natural compounds” [1], δ-lactones appear in a lower percentage, especially as fragrant components in fruits and essential oils, while macrocyclic lactones show up in a small amount, but in numerous active compounds. Lactones are present in pheromones, macrolide antibiotics [6], and numerous naturally occurring lactones that are also extensively studied for their cytotoxic activity against cancers, and for their anti-bacterial and anti-fungal activity [7]. Though prostaglandins were found in numerous marine organisms in the 1970s and reviewed recently [8], prostaglandin lactones were discovered only in the molluscs [9] nearly 14 years after the synthesis of the first prostaglandin lactones, PGF_2α_-1,9-lactone **1** and PGF_2α_-1,15-lactone **2** (Figure 1) [10].

The first prostaglandin lactones found in a marine nudibranch mollusc called *Tethys fimbria* were: PGE_2_-1,15-lactone **3**, PGE_3_-1,15-lactone **4,** and PGE_3_-1,15-lactone-11-acetate **5** [9]. Then, other prostaglandin analogs were found in the same mollusc [11,12,13,14], such as: PGF_2_-1,15-lactone **6**, PGF_3_-1,15-lactone **7**, PGF_3_-1,15-lactone-11-acetate **8**, PGA_2_-1,15-lactone **9**, and PGA_3_-1,15-lactone **10** (Figure 2). Their role in different organs of the mollusc in the reproduction cycle were studied and the studies provided a mechanism by which prostaglandin can be stored and released, when needed [11]. A detailed PG-lactone pathway in *Tethys fimbria* was presented in a review paper [15].

A recent review paper discusses the wide spread of prostaglandins in marine organisms [8], and it seems that, among the prostaglandin lactones, only 1,15 lactones have been found in these organisms so far. 1,9-Lactones and 1,11-Lactones have not been observed. However, they show up a lot in the synthesis of prostaglandins and their analogs. Other structures of δ-lactones and γ-lactones are used to obtain key intermediates in the sequence of prostaglandin synthesis, especially in the Corey procedure. Hence, lactones are of great interest for the researchers. Many methods for obtaining δ-lactones and γ-lactones were actually developed for and in the synthesis of these key intermediates. The big class of prostaglandins and prostaglandin drugs increased the interest of the research community for lactones for the development of new routes to obtain lactones, as well as for their subsequent applications. To our knowledge, a review of this topic was not encountered in the literature and this paper aims to fill in this gap. The paper aims to give the reader a comprehensive overview of the lactones in the field of the prostaglandins, starting from those of a natural origin, continuing with their synthesis (to gain a better understanding of their metabolic role, and to study their applications as a replacement for the C-1 esters and for the preparation of 5-trans free prostaglandin isomers), progressing to the key δ-lactone and γ-lactone intermediates in the milestone Corey strategy for obtaining all series of prostaglandins, in leukotrienes and isoprostane prostaglandin analogs, and ending with the applications of δ-lactones for getting new 9β-halogeno-prostaglandins and halogeno-intermediates (for fine organic chemistry and future research).

## 2. Lactones in the Synthesis of Prostaglandins and Prostaglandin Analogs

In this paper, we present the synthesis of the lactones encountered in prostaglandin and leukotriene synthesis. We include the synthesis of the milestone δ-lactones and γ-lactones used in the total stereo-controlled Corey procedure for the synthesis of prostaglandins and their analogs, and their isoprostane isomers. The chemical separation of pure enantiomers through dia-stereoisomer salts with optically pure amines or by enzymatical procedures, and also highly enantio-selective catalyzed reactions for obtaining enantiopure compounds are presented in the paper. Finally, we treat the synthesis of 9β-halo-prostaglandins and 9β-halo-cyclopentane intermediates from δ-lactones. These directions and, finally, conclusions are presented in the plan outlined below.

### 2.1. Synthesis of Prostaglandin Lactones

#### 2.1.1. Internal Lactonization of an Acid with a Hydroxyl Group of a Prostaglandin by the Corey-Nicolaou Procedure

Prostaglandin lactones were synthesized first by Corey and Nicolaou [10] through the reaction of the C_1_ acid with the corresponding secondary alcohols 15 or 9 by reaction with triphenylphosphine and 2,2′dipyridyl disulfide. In order to obtain 1,15 anhydride **2**, the 9-OH group was protected as acetate whereas, to obtain 1,9-anhydride, both the 11 and 15-OH groups were protected as tetrahydropyranyl (THP) ether. This “double activation” process consists of first activating the acid by transforming it in an ester while activating the hydroxyl group at the same time (this simultaneously activates both the hydroxy and the carboxylic acid moieties with a single proton transfer) and then performing the cyclization to the anhydride (Scheme 1) to give **11**, which was finally hydrolyzed to PGF_2α_-1,15-anhydride.

The reaction is quite general and was used not only for synthesis of other prostaglandin anhydrides, but also as a highly efficient method for preparing medium to large rings, like brefeldin A, erythronolide B, enterobactin, and (±)-vermiculine [16].

Using the Corey and Nicolaou procedure [10], Bundy et al. obtained a series of PGD_2_ compounds as well as PGD_2_-1,9 and 1,15 anhydrides, as follows: from PGF_2α_-15-THP ether [17], they first obtained PGF_2α_ 1,9-lactone-15THP, and then oxidized it with the Jones reagent to PGD_2α_ 1,9-lactone-15-THP **14** and, finally, hydrolyzed it to **15** (Scheme 2).

From 15-methyl-PFG_2α_ the l,9-lactones **16** and **17** (Scheme 3).

For the synthesis of PGD_2_-1,15-lactone, due to the instability of the compound, PGF_2α_ was first transformed into 9,11-butylboronate, lactonized with 2,2′-dipyridyl disulfide/Ph_3_P, the boronate ester was deprotected, and then a selective silylation of 11-OH was performed, followed by the THP protection of 9-OH, the deprotection of the TBDMS group, the Jones oxidation of 11-OH, and, finally, by the deprotection of the THP group. Fortunately, the PGD_2_-1,15-lactone was isolated by crystallization because it was not resistant to purification on silica (neutral or acid washed) (Scheme 4).

Prostaglandin D_2_-1,9-lactones and 1,15-lactones were studied for their platelet inhibiting activity, in comparison with PGD_2_ and its other derivatives. Only 9β- and 9-deoxy-PGD_2_ were more active than PGD_2_. All other synthesized compounds were less active than PGD_2_. A few patents were obtained on the synthesis of prostaglandin 1,9-lactones [18], prostaglandin 1,11-lactones [19], and prostaglandin 1,15-lactones [20,21].

1,15-Lactones of the prostaglandins used in the treatment of glaucoma and the reduction of ocular pressure, fluprostenol, latanoprost, and cloprostenol were synthesized by the same procedure, starting from 9-TBDMS-fluprostenol or cloprostenol acids and 9,11-di-ethoxyethyl-latanoprost acid, which was followed by the deprotection of the protecting groups (Figure 3) [22,23]. In the bovine corneal tissue, anhydrides **24** and **26** hydrolyzed to the free acids fluprostenol and latanoprost at the same rate as the isopropyl esters, showing that the anhydrides could be formulated into ophthalmic solutions as standard isopropyl esters for reducing the intraocular pressure [24]. 13,14-Hydrogenated latanoprost-1,15-lactone and fluprostenol-1,15-lactones are also mentioned in a patent [24]. Prostaglandin F_2α_-1,11-lactone was synthesized by the Corey-Nicolaou method and also studied for the reduction of intraocular pressure [25], showing a marked increase duration of activity and an increased reduction of intraocular pressure, by comparison with PGF_2α_. In addition, the associated hyperemia also decreased.

The 1,9 lactones of prostaglandins used in the reduction of intraocular pressure were used as a step for obtaining the corresponding prostaglandins substantially free of the 5,6-trans isomer (Scheme 5). Latanoprost, isopropyl unoprostone, travoprost, and tafluprost, and their free acid forms are not solids to be purified by crystallization. The patent is based on the fact that the 1,9-lactones of travoprost, tafluprost, latanoprost, bimatoprost, and unoprostone [26] are crystallized compounds and, by isolation and crystallization of 11,15- or 11-unprotected hydroxyl lactones, the ratio of 5,6-trans isomer obtained is below 0.1%. The lactonizations were performed by the Corey-Nicolaou procedure (entries 1, 6, 8, and 9 in Table 1), but other methods, described below, were also used for the lactonization.

In the case of unoprostone (entry 9), lactone **28e** was first deprotected from 15-*O*-TBDMS, and then oxidized to the 15-keto compound. 11-*O*-THP was deprotected to the free acid and, finally, esterified to isopropyl ester **30e** [26].

Two more efficient disulfide reagents, **31** and **32** (Figure 4) [27], were used for macrolactonization, such as for example **32** in the synthesis of *erythronolide B* (aglycone of the important antibiotic *erythromycin B*), but they were not utilized in prostaglandin lactonization.

#### 2.1.2. Internal Lactonization of an Acid with a Hydroxyl Group by the Mixed Anhydride (Yamaguchi and Coworkers) Procedure

The procedure is described as a mild and rapid esterification, performed in two steps [28]. For prostaglandins (protected at 11, 15-OH as THP, for example), in the first step, the C_1_-acid is transformed into a mixed anhydride with a good leaving group by the reaction with 2,4,6-trichlorobenzoyl chloride in the presence of triethylamine as the base. In the second step, the mixed anhydride is internally alcoholized with the free 9-OH group in the presence of 4-dimethylaminopyridine to give 1,9-lactone in a good yield (82.4%, entry 4, Table 1) [26]. Using benzoyl chloride instead of 2,4,6-trichlorobenzoyl chloride gave the same lactone in a little smaller yield (80.3%, entry 2, Table 1). With 11,15-OTES, the yield was lower (61.6%, entry 3), likely due to the deprotection of triethyl silyl during the reaction. Trifluoroacetic anhydride was also used for obtaining 1,9 prostaglandin lactones, but “the strong acidic conditions (CF_3_COOH) employed made this method not a very attractive one” [29].

#### 2.1.3. Internal Lactonization of an Acid with a Hydroxyl Group by the Shiina Procedure

The Shiina procedure [30,31,32] uses substituted benzoic anhydrides for obtaining mixed anhydrides, in the presence of a base (Et_3_N) and a Lewis acid or a base (DMAP, 4-pyrrolidinylpyridine (PPY), or 4-(dimethylamino)pyridine-*N*-oxide (DMAPO)) for the second step, and the reaction of mixed anhydrides with alcohols to give the alkyl esters. Though many substituted benzoic anhydrides were studied in detail [30,32], 2-methyl-6-nitrobenzoic anhydride (MNBA) gave the best results not only in intermolecular esterification, but also in internal esterification with the formation of a macro-lactone [32,33]. The reaction needs nearly equimolar amounts of carboxylic acids and alcohols, about 1.1 equivalents of MNBA and Et_3_N and catalytic amounts of DMAP (10 mol %), which proceeds smoothly at room temperature, without the isolation of the mixed anhydride, with high chemo-selectivity and in excellent yields. The method was applied for the synthesis of PGJ_2_-lactones and PGJ_3_-1,15-lactones and their analogs, including 10-Cl substituted lactones (Scheme 6). In these applications, 1.4 equiv. MNBA, 6 equiv. DMAP, CH_2_Cl_2_ as solvent were used for 12–17 h at 25 °C to obtain the 1,15-lactones **34a**–**34h** starting from the free acids **33**, and also *epi*-**34a**, in moderate to good yields (Table 2). The authors also presented in this paper a shorter and more efficient olefin-metathesis-based total synthesis of the free acids of Δ^12^-PGJ_2_ and of Δ^12^-PGJ_3_, **33**. It is important to mention that the Δ^12^-PGJ_3_-lactone **34a**, **34f** and Δ^12^-PGJ_2_-lactone **34i** were the most promising cytotoxic agents against a variety of cancer cell lines (other compounds synthesized were also active on cancer cell lines [34,35]) and their activity was found to be more potent than that of parent acids or open esters. The same observation is valid for the other lactones, mentioned in Table 2. This observation guided the authors to synthesize dimers, trimers, and tetramers of lactones **34i**–**34k**. If for the lactones **34i**–**34l**, pseudo high dilution macro-lactonization conditions (MNBA, DMAP, ca. 0.8 mM concentration) were used, under modified macro-lactonization conditions [MNBA (1.5 equiv.), Et_3_N (2 equiv.), DMAP (0.1 equiv.), ca. 10 mM concentration, CH_2_Cl_2_, 25 °C, 12 h)] dimeric, trimeric, and tetrameric macro-lactones were produced simultaneously in the same reaction (**34i**: 27%, 10%, 9%; **34j**: 20%, 8%, 4%; **34k**: 19%, 9%, 8%; **34l**: 22%, 14%, 8%). Dimeric macrocyclic lactones showed stronger cyto-toxicities than their monomeric counterparts and the corresponding trimer and tetramer proved significantly less active (or not active) against tested cancer cell lines, especially for **34l** polymers [35].

#### 2.1.4. Internal Lactonization of an Acid with a Hydroxyl Group by the Mukaiyama Procedure

The Mukaiyama procedure initially used 1-methyl-2-chloropyridinium iodide and Et_3_N as the base to form an activated intermediate of the acid, as in the Corey procedure, which then reacts with the hydroxyl group to give the corresponding lactone. Because of the secondary reactions due to the decomposition of pyridinium salts by the attack of triethylamine on the 1-methyl group or the substitution of the 2-chloro atom. Other new efficient reagents for lactonization have been developed, like the stable pyridinium salt, known as 2-chloro-6-methyl-l,3-diphenylpyridinium tetrafluoroborate. This reagent was used for the lactonization of PGF_2α_-9,11-bis-THP to PGF_2α_-9,11-bis-THP-1,15-lactone **36** in 91% yield (Scheme 7) [36,37] and gave the 2-keto-pyridine by-product **38** in the reaction.

The deprotection of the THP-groups gave lactone **37** at a 79% yield. By the same procedure, the unprotected PGF_2α_ was selectively lactonized to PGF_2α_-1,9-lactone **1** with only 4% PGF_2α_-1,15-lactone **37** [36].

Though there are many efficient methods for macro-lactonization to obtain complex natural products [37,38,39,40], these found no applicability in obtaining prostaglandin lactones.

#### 2.1.5. Internal Lactonization by Ring-Closing Metathesis

In the previous methods, prostaglandin lactones were synthesized by using an unprotected or a conveniently protected prostaglandin. In the ring closing metathesis (RCM), the α-side chain is built on an intermediate, which contains dialkene or dialkyne groups in the molecule. In this case, the intermediate is obtained by the “three-component coupling” method [41,42], which is a variant of the most elegant and efficient method for building the prostaglandin skeleton. Fürstner et al. [43,44] employed this method to get the starting alkyne key intermediate **39**, which was esterified at the hydroxyl from the ω-side chain with 5-heptynoic acid by using diisopropylcarbodiimide and catalytic amounts of 4-dimethylaminopyridine (DMAP) (Scheme 8). The resulting dialkyne ester **40** was used in the macrocyclization via the ring-closing alkyne metathesis in the presence of the catalyst formed in situ from Mo(N*t*BuAr)_3_ and CH_2_Cl_2_ in toluene at 80 °C, giving the alkyl-lactone **41** in 68–73% for catalyst with Ar = 3,5-dimethylpyridine and in a better 77% yield with Ar = 4-fluoropyridine. Both catalysts were very reactive for the alkynes and, at the same time, highly tolerant not only toward the double bond in **40**, but also toward other chemical groups [45]. It is worth mentioning that no racemization was observed in the lactonization or in the steps before RCM. The triple bond was cleanly hydrogenated in the presence of the Lindlar catalyst to give 11-TBDMS-PGE_2_-1,15-lactone **42**, which was desilylated with aqueous HF in acetonitrile at a good yield to PGE_2_-1,15-lactone **43**. All steps from **39** to **43** were performed in very good yields and represent a good method to obtain PGE_2_-1,15-lactone and also the corresponding lactones F_2α_ and A_2_. By the same sequences, 15-***epi*-43** was synthesized. In the RCM reaction of ***epi*-40** to ***epi*-41**, both catalysts, mentioned before, gave better yields: 81% and, respectively, 87%.

By this method, other prostaglandin lactones were synthesized starting from the corresponding ester compounds with 9-undecynoic acid and 2-(hex-4-yn-1-yloxy)phenol, an enlarged (ring-expanded) lactone **44**, and a lactone with a salicilic acid fragment in the α-side chain, **45** (Figure 5). In the RCM, the best results were obtained with the catalyst Mo[NtNu(4-FPy)]_3_, 63% for alkyne-lactone analog of **44** and, respectively, 86% for corresponding alkyne analog of **45**, as in the previous examples [45].

RCM with the diene **47** was applied to get the 11,15-protected PGF_2α_ travoprost 1,9-lactone **48** in which the Grubb’s catalyst was used (Scheme 9), but the 16% yield mentioned was low [26]. The deprotection of both silyl ether groups was then performed at an 80% yield.

To find very good yields (Table 3) for obtaining PGF_2α_-1,9-lactone analogs, the same sequence of reactions (**50**→**52**→**53**→**54**→**55**→**56**, Scheme 10) are mentioned in the literature [46]. We would like to point out that the epi-**53** alcohol is transformed into the same diene ester **54** by a Mitsunobu reaction with inversion of the configuration to the C-9 carbon atom. The yields of all the steps from **50** to **56** and **57** are given in Table 3. A better yield (93.3%) of **53** was obtained by the selective reduction of ketone **52a** with L-Selectride.

The reaction conditions used in Reference [26] are the same as those mentioned in Reference [46]. However, there is a discrepancy in the reported values for the yield of the RCM reaction (16% in the first paper, respectively, and 75.0% in the second. We assume the second one is correct).

RCM was used with the scope to build the key lactone intermediate **59** for the synthesis of *cis*-5-F_2_-isoprostane starting from the cyclobutene intermediate **58** (X = H) (Scheme 11), readily obtained by the esterification of **63**, and ethylene in the presence of Grubbs’ catalyst **61**. However, the yield was disappointingly low (37%) (Scheme 11). In this case, there is first a ring opening of the cyclobutene ring with ethylene followed by the RCM of the formed diene, which is similar to **64** (Scheme 12), with the double bond of the 6-heptenoic ester from **58**, giving the lactone isomers *Z*-**59** and its isomer *E*-**60** in a ratio of 3:1. With **58** (X = O), no lactones were obtained [47], and an ingenious study was performed to obtain the intermediates **66** and **69** with an α-side chain by the RCM method.

By a stepwise meta-thesis of the cyclobutene compound **63** with ethylene and Grubb’s catalyst **61**, followed by the cross-meta-thesis of the resulted diene **64** with α,β-unsaturated ester **65** in the presence of Hoveyda-Grubbs’ catalyst **62** (Scheme 11), the key intermediate **66** was obtained at a 77% yield, as a single regio-isomer and *E*-stereoisomer in a single reaction vessel (Scheme 12) [47].

In the case of the cyclobutene compound **67**, though the yield of **68** is quantitative, for optimal yield, the next cross-metathesis of this diene with **65** was run to partially convert it to **69**, and the unreacted **68** was recycled (3×) in cross-metathesis to give the desired *E*-key intermediate **69** in an overall yield of 68% (Scheme 13) [47].

Though the lactonization of **58** with the α,β-unsaturated ester **65** did not proceed, the solution to study the mechanism of the metathesis of cyclobutene compounds **63** and **67** with ethylene and **65** to obtain the compounds **66** and **69** as a single regio-isomer and *E*-stereoisomer in good yields in a single vessel is remarkable. In fact, three reactions were avoided: the esterification of 9-hydroxyl, the lactonization in Scheme 11, and the final esterification of the C_1_-acid. Moreover, the α,β-unsaturated ester **65** was used free, and not linked to the core fragment used in the metathesis reaction.

RCM with other ruthenium catalysts was used for directly building the α-side chain of PGF_2α_ or PGE_2_, at a 59% and 51% yield [48,49], without the protection of the hydroxyl groups, without the esterification of 9-hydroxyl or 15-hydroxyl group and without using a prostaglandin lactone as a key building block, as mentioned above. RCM was also used to form the ω-side chain of isocarbacyclin and its 15R-TIC analog [50] with the goal to overcome the selective, but difficult, reduction of the 15-keto group by using an alkene intermediate with an enantiomerically pure 15-hydroxyl group **71**. RCM was realized with the advanced key intermediates **70** and **71** in the presence of Grubbs’ second-generation catalyst in good 82–90% yields (Scheme 14). The same RCM reaction was performed with the corresponding OH-protected Corey lactone. The corresponding bicycloketones for building the ω-side chain were obtained in 84% and 86% yield, and then used in the synthesis of PGF_2α_ and carbacyclin [50].

For RCM, many complex ruthenium catalyst [51] and others were synthesized, which could improve the yields of the reactions mentioned above, so the variant for synthesis of prostaglandin lactones as prodrugs or as a step for obtaining the prostaglandins analogs should be taken into consideration.

#### 2.1.6. Other Methods for Building Prostaglandin or Eicosanoid Lactones

1,5-Lactones of prostaglandins were also synthesized. For example, by haloeterification with iodine/NaHCO_3_ or NBS/NaHCO_3_, PGF_2α_ was transformed into 5-hydroxy-6-iod (or 6-bromo)-PGF_1α_-1,5 lactone **74** and then used in a sequence of reactions to obtain first the ∆^6^-Z-compound **75** and then the 7-keto-prostacyclin analog [52] (Scheme 15) with platelet aggregation-inhibition activity weaker than that of prostacyclin, but with higher stability.

1,5-δ-Lactones were also found in the eicosanoids. For example, cyclo-EC, **77**, was first observed to be formed in the synthesis and NMR analysis of EC, **76**, [53]. Then it was synthesized in 65% yield by the acid (on silica gel) catalysis ring opening of the epoxide ring by the 1-carboxyl group (Scheme 16) [54]. It has been discovered that Cyclo-EC **77** has the most anti-inflammatory activity in comparison with that of 5,6-epoxyisoprostane **76** and isoprostane **78** as inhibitors of proinflammatory cytokines IL-6 and IL-12 secretion.

Later, other 1,5-lactones and 1,5-lactames and their analogs **79a**–**79c**, **80**, respectively, **81a**–**81c** were synthesized [55] and their anti-inflammatory activities were established. Lactone **77** was the most active one, while the 1,5-lactames **81a**, **81b**, and **81d** retained high anti-inflammatory activity (Figure 6). The homologated analogs **79b**, **79c**, and compound **80** with a *Z*-linked α-side chain were inactive.

5,6-Epoxy-eicosatrienoic acid (5,6-EET, **82**) is formed by cytochrome P450s oxidation of arachidonic acid in mammalians and is spontaneously cyclized, as EC 76, to the stable 1,5-lactone 5,6-DHTL, **83**. Then it was found that lactone **77** and **5,6-DHTL** (**83**) are hydrolyzed in mammalian paraoxonases, which is a family of esterases, to the corresponding free acids (Scheme 17) [56].

(R)- and (S)-6,7-dhydro-5-HETE lactones **89** were synthesized by linking a key chiral aldehyde intermediate **85** with the yield generated in situ from the triphenylphosphonium bromide **86** in a *cis*-selective Wittig reaction with the generation of the triene **87a** in a 68% yield. The deprotection of the trityl group (71%), followed by the oxidation to 1-carboxylic acid (73%), and the final acid cleavage of the MOM ether, gave directly the (R)-6,7-dihydro-5-HETE lactone **89** at a 92% yield (Scheme 18) [57]. (S)-isomer **89** was obtained by the same sequence of reactions starting from the enantiomer **85**.

Though the prostaglandin lactones were discovered in nature only in the molluscs and the eicosanoid lactones were found in the metabolic pathways of unsaturated fatty acids to prostaglandins. The synthetic effort to obtain these compounds and plenty of their analogs was impressive. In many cases, the synthesis of 1,15-, 1,11-, and 1,9-lactones of prostaglandins and their analogs gave more active prodrugs than the parent prostaglandins. Sometimes the crystallized lactones were used in the synthesis sequence to obtain free 5-*trans* isomer (<0.1%) prostaglandin compounds, which were, otherwise, difficult to obtain by other processes. In conclusion, efficient methods for their synthesis were developed, and further applied to other classes of lactones.

### 2.2. Synthesis of γ-Lactone Intermediates in the Sequence for Obtaining Prostaglandin Analogues

#### 2.2.1. Synthesis of Corey γ-Lactone by the Most Used Methods

δ- and γ-lactones are strongly linked to the convergent total stereo-selective Corey synthesis of prostaglandins in which three key intermediates are independently synthesized: one for building the cyclopentane skeleton and two for building the α-side and ω-side chains in the final stages of the sequence. This was a pioneer procedure, which soon grew into an efficient method for the synthesis of the core γ-lactone **95** with four chiral carbon atoms and the substituents in the right positions for linking the α-side and ω-side chains and for 9α,11α-hydroxyls of the prostaglandin skeleton (Scheme 19). The Corey γ-lactone **95**, as one of the three key intermediates, *was used for the synthesis of the three series of prostaglandins: PG_1_, PG_2_, and PG_3_, becoming a general method for their synthesis.* A lot of syntheses for obtaining racemic and mainly optically active **95** were discovered, starting from different racemic or optically active raw materials. Some of them were improved and applied to the large-scale synthesis of this key intermediate. One of the first syntheses is presented in Scheme 19. The first step of the sequence was many times improved from the first Diels-Alder reaction of diene **90** (OMe) with 2-chloro-acrylonitrile, as a ketene equivalent, catalyzed by Cu(BF_4_)_2_, to give racemic **93** (80%) [58], to the highly enantioselective Diels-Alder of **90** with acrylate of 8-phenylmenthol (**91a**) (AlCl_3_ catalyst), giving **92a** in 89% yield and 97:3 dia-stereoselectivity [59], and then the catalyzed reaction of benzyloxymethylcyclopentadiene **90** with 3-acyloyl-1,3-oxazolidin-2-one **91b** (10 mol % catalyst **96**), which gave the chiral Diels-Alder adduct **92b** in a 94% yield and 96% *ee* [60] (Other chiral acrylates were also used [61]). For the enantioselective Diels-Alder reactions, other catalysts (like **97**) [62] were then synthesized and gave even greater yields, such as 99% *ee* (>99% *exo*) in the reaction of **90** with 2-bromoacryloin aldehyde [63]. The next steps to the bicyclic ketone **93** were realized in an 83% yield. The transformation of **93** into iodo-γ-lactone **94** was realized in an 89% yield by Baeyer-Villiger oxidation, opening of the lactone to the unstable acid and iodolactonization. The removal of the iodine from the Corey lactone **94a** was realized at a 99% yield with tributyltin hydride in benzene at 25 °C (initiation with azobisisobutyronitrile) [64]. The next steps to achieve the needed protection of the primary and secondary groups are easily realized by the well-known protection-deprotection methods.

It is worth mentioning that, in this procedure, the γ-lactone group was constructed by opening a δ-lactone group to acid and then by the γ-lactonization by iodohalogenation of the double bond, with the halogen being removed at a near quantitative yield to the Corey γ-lactone **94a** (de-iodinated).

A different but also largely applied approach to efficiently obtain the Corey Lactone **103** is presented in Scheme 20. The sequence started from norbornadiene and gave **100** in three steps in good yields: Prins reaction, Jones oxidation to **99**, and HCl (also HBr and HI) opening of the cyclopropane ring to **100** [65,66]. The next steps, **100** to **102** (Baeyer-Villiger oxidation, reduction of carboxyl and protection of the alcohol) were performed in good yields [67,68]. Another efficient route, **100**→**104**→**105**→**106**→**102a**, which consists of the protection of the keto group as ethyleneketal (91.6%), the borane reduction of the acid group followed by the removal of the protecting group (92% on two steps) and by the Baeyer-Villiger oxidation (87%), made the sequence norbornene→**102a** one of the most used methods, together with that mentioned above, for obtaining the δ-lactone **102** and the Corey γ-lactone **103** [69]. In the sequences, optically active intermediates were obtained by cleaving the racemic compound into enantiomers by the dia-stereo-isomer salts of the acid compounds **99**, **100**, and **104** (in red in Scheme 20) with optically active amines, like (**−**)-1-phenylethylamine, ephedrine, quinine (see, for example, the separation of **99** and **104** with (**−**)-1-phenylethylamine, ephedrine [66,70], of the Corey lactone **103** (R = TBDMS or Et_3_Si) with a chiral non-amine scaffold, (**−**)-(1R,4R,5S)-4-hydroxy-6,6-dimethyl-3-oxabicyclo [3.1.0]hexane-2-one [71]), or by reacting **95** (R^1^ = *p*PhBz) with anhydrides and then with optically active bases [(**−**)-(1*S*,2*R*)-ephedrine, (**−**)-(*S*)-1-phenylethylamine, quinine) [72].

In comparison with the previous procedure, in this method, the δ-lactone is built by an insertion of an oxygen atom, and the transformation of a δ-lactone into a γ-lactone is realized by an intramolecular SN_2_ substitution of the chlorine atom by the carboxylate anion.

Another route, studied in depth, and now becoming widely used in the synthesis of the racemic Corey lactone **95**, starts from the cyclopentadiene, which was used in the Diels-Alder cycloaddition with dichloroketene to obtain adduct **105** [69,70,71,72,73,74] (Scheme 21). Then, the dechlorination and Baeyer-Villiger oxidation give the γ-lactone *cis*-2-oxabicyclo[3.3.0]oct-6-en-3-one **107** in very good yield (~72%). **107** has two carbon atoms with the stereochemistry required for the structure of Corey γ-lactone **95**. The other two stereochemical requirements are introduced by transforming the double bond of **107** into a *cis*-regiospecific Prins reaction to give the acetylated lactone **108**, together with small amounts of monoacetylated products at the primary or secondary alcohol groups [75,76]. In the next transesterification reaction (MeOH/MeONa), **108** and the monoacetylated compounds are hydrolyzed to the racemic **109**. The next protection of the primary or the secondary alcohol group is realized as usual.

In order to obtain the optically active Corey lactone **107**, compound **(±)-****107** was transformed into the hydroxy-acid (base hydrolysis, acidify to pH 3.5–4, extract the hydroxyacid), which was then reacted with (+)-α-methylbenzylamine and the pure diastereomer salt (obtained by fractional crystallization) was treated first with base (to remove the optically active amine), then with acid, and extracted to give **(−)-107** [77,78] (Scheme 22). A similar separation of the enantiomers of the **107-acid** is also described in recent patents [79,80].

The optically active lactone **107** was also obtained by another reaction sequence from cyclopentadiene→**110**→**(****−)-107**, in which alkene **110** was asymmetrically reduced with (+)-di-3-pinanylborane, followed by the alkaline hydrogen peroxide oxidation to yield the hydroxy ester **(****−)**-**111** at a 45% yield (Scheme 23) [78].

Due to its high yield sequence and the cheap starting compounds, this method was improved and is used extensively for obtaining the Corey lactone **107**. Recent reviews on the Corey lactone (with different substitutions of the hydroxyl groups) in relation to its use in prostaglandin syntheses of drugs, analogs in different clinical phases, or other analogs are referenced in many papers [81,82,83,84,85], but no in-depth review paper focusing only on the procedures for the synthesis of Corey lactones was found in the literature.

#### 2.2.2. The Improvements in the Synthesis of Corey Lactone

Though the above methods were largely used at high yields, the researchers more deeply studied some of the steps of the sequences in order to simplify them or reduce their number, to make one-pot synthesis for a number of steps, to change the reagents, to change the solvents, and to use stereo-controlled reactions for some steps. The main achievements of this line of research are presented below.

An improvement in the preparation of the Corey lactone acid **113** was realized by performing the Baeyer-Villiger oxidation and opening the δ-lactone **101** followed by the ring closure to the γ-lactone ring of **113** in a one-pot synthesis (for two steps) with potassium persulfate (OXONE) in the presence or absence of 4-butylammonium bromide as a phase transfer catalyst. The yield claimed is high (not given) (Scheme 24) [86].

Another improvement in the transformation of δ-lactone **114** into γ-lactone **109** is presented in Scheme 25. During the study of the ring opening of the δ-lactone **114** by acid methanolysis, the chloroester **115** was obtained at a quantitative yield (TsOH or Amberlit IRC 50W × 2, reflux). The benzoate chloroester **117** was also obtained at a quantitative yield by methanolysis in CH_2_Cl_2_-MeOH (1:1) without the deprotection of the benzoate group after two days at rt (TsOH) (See Section 2.4) [87,88]. After the protection of the secondary alcohol with an ether group, the base hydrolysis of the esters (methyl and benzoate) and the internal closure of the furane ring, the γ-lactones **120** were obtained at a 92–93% yield for R^2^ = TBDMS, 93% for R^2^ = THP, 90% for R^2^ = Tr, 95.5% for R^2^ = Tr, and with *tert*-butyldimethylsilyl (TBDMS) protected primary hydroxyl. The base hydrolysis (NaOH in CH_2_Cl_2_-MeOH-water, rt, overnight) of the chloro-ester **115** gave the crystallized compound **109** in a yield greater than an 80% yield (unoptimized). Hence, the transformation of the unprotected δ-lactone **114** and of the ester protected lactones **116** (R = Bz, p-nitro-benzoate, acetate) into the γ-lactones **109** and **120** is an efficient procedure, providing a good alternative to those mentioned previously, which use a base with great excess of H_2_O_2_ for this transformation.

The racemic bis-chlorinated cyclobutanone **105** was transformed into the enantiomer γ-lactone **(+)-****121** by a stereo-controlled organometalic Baeyer-Villiger oxidation at a 46% yield and in 99% *ee* with a good selectivity factor, 118, in the presence of the phosphoric acid catalyst **(*R*)-124** (Scheme 26a). With the enantiomer catalyst **(*S*)-124**, the desired optically active lactone **(−)-121** was obtained at a 45% yield and 95% *ee*. Other dichlorocyclobutanones linked to different single or fused rings were also stereo-selectively transformed into the corresponding γ-lactones at a 30–47% yield and 90–99% *ee*. The following two steps, consisting of the reductive dechlorination with Zn/NH_4_Cl and the Prins reaction with paraformaldehyde in formic acid and H_2_SO_4_ as a catalyst, gave the optically Corey γ-lactone **(−)-109** at a quantitative yield for the first step and, respectively, at a 79% yield for the second step (Scheme 26b) [89,90].

The Corey lactone **95** is oxidized to aldehyde and used for building the ω-side chain by an *E*-HEW selective olefination. The aldehyde from **95** can be protected as dimethyl acetal and used in this form in the next steps for building the α-side chain, as it is usually done when the primary alcohol is protected. The dimethyl acetal (rarely diethyl acetal), which is an ethylene ketal group, was used for protecting the aldehyde group of the intermediate **125** in the next steps for building the α-side chain (Scheme 27) [91].

The aldehyde **127**, obtained by the oxidation of the lactone **102a** (R = H, Scheme 20), was ketalyzed by alcoholysis with diols (X = O) and dithiols (X = S) in acid catalysis, concomitant with the opening of the δ-lactone ring, to give the halogenated compounds **128**. Then the base hydrolysis of ester (KOH in THF-water) gave the γ-lactone aldehydes **129** protected as ketals, where X = O and X = S (*n* = 0 and 1 in both cases) (Scheme 28) [92]. The protection of the hydroxyl group of compounds **128** as ether (THP, TBDMS) gave compounds **130** with the corresponding protecting groups and simplified the entire preparation procedure. Compounds **129** were also protected with ether or ester groups. It is also worth pointing out that compounds **129**, either racemic or enantiomerically pure, are obtained in a crystalized form and they are stable for years and represent good key intermediates for the PG synthesis.

#### 2.2.3. Synthesis of the Corey Lactone by Other Promising Methods

Many other methods for obtaining optically active Corey lactone, with different protections of the primary and secondary alcohol, were studied, and in what follows we will review the most promising alternative methods developed after 1990. Some of these methods were discussed in other review papers [83], but we will present them here in a modified form (the differences are reflected in the Schemas below), to show evidence of those methods developed at high yields from different raw materials, the applications of regio-selective and enantioselective reactions, the enzyme steps for obtaining optically active compounds, and the ingenious strategies for lactones, with a potential value for future applications.

Likely, the most promising method is the one recently developed by Aggarwal and coll. (Scheme 29) [93]. An aldol condensation of succinaldehyde with catalytic amounts of L-proline and dibenzylammonium trifluoroacetate (DBA) (2% each) gave the key bicyclic enal **133** at a 14% yield. The first step consists of an L-proline catalyzed enantioselective intermolecular aldol reaction between two molecules of succinaldehyde to create the intermediate **132** and then DBA catalyzed the second intramolecular aldol reaction and dehydration of **132** to the γ-lactone **133** with 98:2 *er*. These steps were improved later to 29% of isolated **133** [94]. The intermediate **133** was then transformed into the more stable OMe ether **134** or it was oxidized to the corresponding lactone. Though the yield for obtaining the key intermediate **134** is low, the starting reagents are cheap and only 4 to 5 steps are needed to obtain PGF_2α_, latanoprost, and bimatoprost in good yields, which recommends the method for future efficient synthesis of prostaglandin analogs [93,94]. The key intermediate **134** or the corresponding oxidized lactone were used in the conjugate addition procedure to obtain alfaprostol and PGF_2α_ [93,94,95]. The method is expected to greatly reduce the costs of the prostaglandins and their already developed drug analogs.

An efficient one-pot synthesis of the Corey lactone **109** was realized by the pivot domino Michael/Michael reactions in the presence of the catalyst **(*R*)-137** and of *p*-nitrophenol in *i*-PrOH, in which the three contiguous stereo-genic centers, C_3a_, C_4_, and C_5_, and the groups linked to C_3a_ and C_4_, needed for Corey γ-lactone structure, are introduced in the intermediate **138** (Scheme 30) [96]. In the next step, the high dia-stereo-selective reduction of the ketone with the bulky hydride LiAlH(O*^t^*Bu)_3_ and the reduction of the aldehyde group takes place concomitantly with lactone formation. The construction of the Corey lactone is finalized by the conversion of the Si-Ph bond into an Si-F bond, which is followed by the oxidation of the C-SiMe_2_F to C-OH with the same stereo-configuration. The whole sequence, starting from the commercially available racemic starting compounds: 3-(dimethylphenylsilyl)propenal (**135**) and ethyl 4-oxo-2-pentenoate (**136**) and the reagents, is realized in 152 min and gives the lactone **109** as a single isomer (>99% *ee*) at a 50% yield. The only chiral compound used in the sequence was the catalyst **(*R*)-137**. The Corey lactone **109** was also synthesized via the five operation in a total yield of 58% with four purifications and in a nearly optically pure form [97].

The intramolecular RCM reaction of the acetoacetate of dialkene **142** (made in situ) in the presence of the second generation Grubbs’ catalyst **145** gave the cyclopentene intermediate **143** (procedure *b*). By procedure *a*, with the first-generation Grubbs’ catalyst **145a**, the same yield of **143** was obtained in a two-step reaction. The treatment of **143** with methansulfonyl azide and deacetylation gave **144** at a 52% yield. A one-pot reaction from **142** gave **144** in a similar yield. The enantioselective intramolecular C-H insertion in the presence of chiral rhodium catalyst ***S*-146**, gave the lactone **(****−)-107** at a 73% yield and 91% *ee*. The chiral *R*-catalyst **146** gave the opposite oxabicyclic antipode lactone **(+)-****107** at a 59% yield (89% *ee*) (Scheme 31) [98]. It is worth mentioning that the chirality of both enantiomers was introduced in the last step with the chiral catalysts ***S*****-146**, respectively, ***R*-146**.

Synthesis of **(−)-107** was also realized at a good yield by starting from the racemic **(±)-147**, which was easily obtained from cyclopentadiene. Both enantiomers of **(−)-** and **(+)-147** were efficiently separated by lipase PS mediated resolution in >99% *ee* and good yields (Scheme 32) [99] and both were next used for their efficient enantio-convergent transformation into the same desired enantiomer (–)-oxabicyclo[3.3.0]oct-6-en-3-one, **(−)-107**. Enantiomer **(+)-147** gave **(−)-107** in an excellent 73% yield, in a two-step reaction: an Eschenmoser reaction with dimethylacetamide dimethyl acetal to **148**, followed by the acid deprotection of both the functional group and the closure of the γ-lactone ring.

The other enantiomer **(−)-147Ac**, separated as acetate in the enzyme resolution, was quantitatively hydrolyzed, protected as TBDMS ether and the cumuloxyether group was removed by a Birch reaction (Na in liq. NH_3_) to give **(−)-149**. This allylic alcohol was transformed then, by the same two reactions used for **(+)-147**: Eschenmoser and acid hydrolysis, into the same compound **(−)-107** at a 67% yield (two steps) (Scheme 32). The procedure opens a new and efficient route for the important key PG intermediate **(−)-107**, by an enantioconvergent synthesis, which uses both enantiomers, separated by the enzyme resolution of **(±)-147**. Other procedures for obtaining optically active mono-protected diols like **147** are also presented in the literature [100,101,102,103,104].

The key prostaglandin intermediate **(−)-107** was also obtained by a sequence of high yield reactions (Scheme 33) [105]: the lipase-mediated de-symmetrization of the *meso*-diol **152** at a quantitative yield, the change of the protection of the hydroxyls (**153**→**154**) (98%), the esterification of the free hydroxyl with phenyl vinyl sulfoxide (100%), the thermolysis of β-sulfoxyethyl ether, and the deprotection of the TBDMS group (60% over 2 steps) and the pyridinium chlorochromate (PCC) oxidation of the lactol to the γ-lactone (79%). This is also a facile route to the PG key intermediate **(−)-107**.

In another procedure, the racemic γ-lactones **164** and **167** were obtained, by a lengthy and inefficient method as that of Aggarwal et coll. [84], starting from the commercially available monobenzyl ether of *cis*-1,4-but-2-enediol **157** (Scheme 34) [106,107]. The reaction with ethyl orthoacetate (10% hydroquinone catalyst) at 140–150 °C gave the protected allyl ketene acetal **158**, which underwent thermal re-arrangement to the unsaturated ester **159** at a good yield (84%). The reduction with DIBAL-H at low temperature (−78 °C, 90%) or with LiAlH_4_ (rt, 91%) followed by the oxidation (rt, 88%), gave the aldehyde **160**. The elongation of the chain with the Grignard reagent obtained from *tert*-butyl α-bromoacetate and magnesium, activated by methyl iodide, or with ethyl diazoacetate in the presence of tin (II) chloride as a catalyst, gave the β-hydroxy ester **161b** and β-keto ester **161a** in good yields. The chemo-selective reduction of **161a** with sodium borohydride in ethanol gave 3-hydroxyester at an 85% yield as a mixture of diastereoisomers. The base hydrolysis of the ester gave the hydroxy acid **162** as a mixture of diastereoisomers, which was bicyclized (with potassium acetate in acetic anhydride at room temperature for 2 h and then in refluxing conditions for 3 h. For the mechanism of bicyclization, see Reference [106]) to **163** (*endo*-*exo* isomers, 3:1) at a 93% yield, and the next Baeyer-Villiger reaction (AcO_3_H, 90%) gave the γ-lactone **164** in the same ratio of inseparable isomers. The following bromohydrins (70%) were possible to separate and each de-hydro-brominated with 1-ethylpiperidine hypophosphite-AIBN to the Corey lactones **167** and 12-endo- **168**. The all-cis lactone **168** is used for the synthesis of isoprostanes (See also Electronic Supplementary Information for [106]). The procedure is mentioned due to its ingenuity, but it does have some drawbacks: the low selectivity for obtaining cyclobutanone **163** as a mixture of endo-exo isomers (3:1), which remains until the final bromohydrines **165** and **166**, and the fact that only racemic compounds are obtained.

#### 2.2.4. Corey Lactones for the Synthesis of Isoprostanes

The Corey lactone is the key intermediate for introducing the cyclopentane fragment for all prostaglandin analogs in the Corey variant synthesis. For the synthesis of isoprostanes (IsoP), a distinct family of prostaglandins characterized mainly by a *cis*-arrangement of the two side chains [108]. An all *cis*-Corey lactone (for ex, **175** in Scheme 34, protected at the secondary alcohol) is required. Its synthesis, which was also presented in Scheme 33, can be found in the literature and a few variants are outlined below.

The all-cis-Corey lactone **175**, which is a key intermediate in the process for obtaining *ent*-5-F_2c_-IsoP and 5-*epi*-*ent*-5-F_2c_-IsoP, was synthesized starting from the isomer tricyclic ketone **169**. First, the optically active carboxylic acid **169** was treated with HCl, as for compound **99**, to give **170** at an 82% yield (99% *ee*) (Scheme 35) [109,110]. Then, two routes to obtain the chloro δ-lactone **173** were followed. In the first route, **170**→**171**→**173**, the Baeyer-Villiger reaction of ketone **170**, gave regio-selectively δ-lactone **171** at an 88% yield and then the chloroformate of the acid was reduced to the chlorolactone alcohol **173** at a 66% yield. In the second route, first, the carboxylic group was reduced by the same procedure at an 85% yield to **172** and then the Baeyer-Villiger oxidation gave lactone **173** at a 63% yield. It should be noted that lactone **173** was obtained in about the same yield. The oxidation and reduction reactions have yields that are almost inverse in the mentioned routes. The bicyclic δ-lactone **173** was protected as *p*-methoxybenzyl (PMB) ether by treatment with *p*-methoxybenzyl tri-chloro-acetimidate in the presence of a catalytic amount of triphenylmethyl tetrafluoroborate at an 87% yield and then hydrolyzed and re-lactonized with LiOH/H_2_O_2_ in THF-water to give the all-*cis* protected Corey lactone **175** at an 82% yield, which is a key intermediate for the synthesis of isoprostanes.

For the synthesis of isoprostanes of type 15-F_3t_, characterized by a cis-correlation of the hydroxyls and of the side chains to the cyclopentane ring, but a trans-correlation between the hydroxyls and the side chains, another strategy was developed by a new route to the key intermediates **177** and **179** (Scheme 36) [111]. δ-Lactone **179**, used for the synthesis of 15-F_3t_-IsoProstane and its 15-epimer [111], was synthesized from the optically active intermediate **(+)-****176**, readily obtained from 1,3-cyclooctadiene in five steps. After the protection of the hydroxyl groups and subsequent ozonolysis of the double bond and reductive work-up, the bicyclic diol **177** was obtained at a 78% yield. The diol was then selectively oxidized in the presence of the efficient catalyst **178** [111,112] for mild oxidative lactonization of 1,4-diols or 1,5-diols **177**, giving the lactone **179** at a 91% yield. The diol intermediate **177** was enzymatically regio-selectively monoacetylated to the hydroxyl linked to the longer chain [113] and was also used for the synthesis of *ent*-7-epi-F_2t_-dihomo-IsoP, 17-F_2t_-dihomo-IsoP, and also of 5-F_3t_-IsoProstane [114].

Other routes for obtaining Corey lactone key intermediates are presented in the literature starting from different raw materials (Figure 7), but these were not applied at a large-scale synthesis (See Reference [83]).

−An optically active compound **180**, (obtained in 14.6% from 2-deoxy-D-ribose, in a nine-step reaction) was used to give the optically active *ent*-Corey lactone in four steps (~37%) [115].−An adduct of 3-carbomethoxy-2-pyrone (3-CMP) with vinylselenide, **181**, was used in a six-step reaction (41%) to give (±)-Corey lactone substituted as Me ether at the primary alcohol and acetyl at the secondary one [116].−An optically active diazo compound **182** was used to give Corey lactone **(−)-****95** (R^1^ = *p*-PhBz) in six steps (~19%) [117].−The optically active compound **183** (obtained in three steps from dimethyl 3-oxoglutarate and glyoxal, followed by enzymatic demethoxycarbonylation) gave **(−)-****95** (R^1^ = Ac) in eight steps [118].

#### 2.2.5. Other Lactonizations for Obtaining Corey γ-Lactone Intermediates of Type **185**

A novel approach to Corey type γ-lactones **185** consists of the reduction of the keto group of the compound **184** with a reducing agent, such as lithium tri-*sec*-butylborohydride, at a low temperature (<−70 °C). In the base conditions of the reaction, the ring of γ-lactone **185** is closed, from the acid and the resulted alcohol, at a >80% yield [119] (Scheme 37). It is interesting that the compound **184** is obtained by the coupling of the ω-side chain to a cyclopentenone intermediate and the following steps are continued by the Corey procedure for building the α-side chain.

#### 2.2.6. δ-Lactones in the Synthesis of Prostaglandins

δ-Lactones **186**, with R = COOH, or ester, CH_2_OH, or protected as ether or ester, aldehyde or protected aldehyde (for ex., dimethyl acetal), are closely linked to the procedure for the synthesis of the corresponding Corey lactones starting from norbornadiene. In this procedure, δ-lactones **186** are only transient intermediates to γ-lactones **187**, as shown in Scheme 38 (See also Scheme 20, Scheme 24, Scheme 25, Scheme 28 and Scheme 35).

A few procedures for obtaining prostaglandins took into account the advantage that aldehyde **186** (R = CHO) could be used directly in the *E*-HEW selective olefination for building the ω-side chain of the enones **188**. The *E*-HEW selective olefination of aldehyde **187** (R = CHO) to enone **189** needs the protection of the secondary alcohol. Therefore, one less step is excluded in the direct olefination of aldehyde **186**. This advantage was used for example in the synthesis of dimoxaprost through the enone **188a** [66,120,121], Cloprostenol, fluprostenol, travoprost through the enones **188b** [66,122,123], PGF_2α_ through the enone **188c** [66], and others. The δ-lactone enones **188** were also used for obtaining 9β-halogenated prostaglandin analogs (See Section 2.4, below) or 9β-halogenated cyclopentane intermediates (See Schemes in Section 2.4 [87,88]). In conclusion, both δ-lactones and γ-lactones are key intermediates for building the core cyclopentane fragment of the prostaglandins and their analogs.

### 2.3. The Use of Enzymes for Obtaining Enantiomerically Pure Key δ-Lactone and γ-Lactone Intermediates for the Synthesis of Prostaglandins

The chemical separation of racemic intermediates by the resolution of an acid intermediate with optically active amines, like *R*-(+)-phenylethylamine. L-ephedrine is efficiently utilized for obtaining optically pure enantiomers of the key intermediates used for the synthesis of optically active prostaglandins and their synthetic analogs. Enzyme biocatalytic transformations were also widely applied to produce almost the same optically active key intermediates. Though intensive research studies have been done for obtaining chiral intermediates for building the ω-side chain, these will not be discussed in this paper. Here, we are interested in the enzyme research paper dealing with obtaining enantiomerically pure δ-lactones and γ-Corey lactones. In principle, the enzyme acetylates selects only one enantiomer of the racemate or hydrolyzes selectively only one acetate of the racemate acetates. This enzymatic separation of enantiomers was applied at different intermediates from the sequences to obtain the final optically active Corey lactone intermediate **(−)-109**.

#### 2.3.1. Selective Enzyme de-Acetalization (Acetate Hydrolysis)

The enzyme resolution of an earlier intermediate in the sequence of synthesis of the Corey lactone has been done by the selective hydrolysis of the acetate group of the enantiomer intermediate **(−)-191**, with porcine pancreatic lipase in aqueous phosphate buffer at rt [124]. The hydrolysis was finished when the ratio of enantiomers became nearly ~1:1 and the enantiomer excess of **(+)-190** surpassed 98% *ee*. During work-up, the acetate of the desired isomer **(+)-191Ac** was separated at an organic phase (ethyl acetate-hexane, 1:1) and the hydrolyzed enantiomer **(−)-191** in an aqueous phase. Therefore, the separation of enantiomeric compounds was easy. The following three steps until the Corey lactone acetate **(−)-103Ac** and *ent*-Corey lactone **(+)-103** are standard (see Scheme 39).

Enzymatic hydrolysis of the acetate was also performed with the racemic Corey compound **194** to separate the enantiomers with lipase PS (Amano, *Pseudomanas sp*.) in 0.1 M phosphate buffer at a pH of 7.6 at 30 °C for 48 h. The target alcohol **(−)**-**195** and the un-hydrolyzed acetate *ent*-**(+)**-**194** were obtained in greater than 99% *ee* and in good yields (Scheme 40) [125].

#### 2.3.2. Selective Enzyme Acetalization

This procedure is mentioned in the literature for the Corey lactone **(±)-109** and for an earlier key intermediate, the γ-lactone **(±)**-**107**.

The racemic Corey lactone **(±)-109** was separated in enantiomers by the selective enzymatic acetylation with lipases of only the wanted enantiomer **(−)-109**, the most important results being obtained with Lipase Amano AK in DCM, using vinyl acetate as an acetate reagent (Scheme 41) [126].

Another acylation was realized by the enzymatic transesterification of the racemic mixture of 5-*p*-phenylbenzoate **(±)-109a** with tributyrin (glycerol tributyrate) in the presence of lipase (triacetylglycerol-acylhydrolase EC 3.1.1.3, type VII from *Candidu cylindracea* from Sigma) [127,128]. In this case, the enantiomer was acylated to the free primary alcohol to give 5-*p*-phenylbenzoate-4-butyryloxymethyl lactone **(+)-109a** (Scheme 42).

The racemic γ-lactone intermediate **107** was efficiently separated into **(−)-107** and its enantiomer **(+)-107**, by enzymatic resolution with lipases: AK, PS, Amano M10, T, AP6, MY, CP, AL, PL266, and PL697 by selective acetylation of the only enantiomer to **(+)-107**, in yields ranging from 91.6% to 98.5% and 90.0% to 99.7% *ee*, with the best results being obtained with lipase AK (98.5% yield, 99.7% *ee*) (Scheme 43) [129].

Enzymes were also used in the microbiological mediated Baeyer-Villiger reaction of the racemic cyclobutanone **(±)-106** to obtain the optically active **(−)-107** at a 40% yield and >97% *ee* (Scheme 44a) [130]. A slightly improved procedure gave a 46% yield [131]. In the reaction with a whole cell culture Actinobacter TD 63, the abnormal” lactone **(+)-195** also resulted in high regioselectivity and in high *ee* selectivity. This unexpected regio-isomer **(+)-195** is important for its use in other applications because it cannot be synthesized selectively by chemical oxidation [132]. Later, regioselectivity of 2-oxo-Δ^3^-4,5,5-trimethylcyclopentenylacetyl-CoA monooxygenase from *Pseudomonas putida* (OTEMO) for the Baeyer-Villiger oxidation of *cis*-bicyclo[3.2.0]hept-2-en-6-one **(±)-106** was studied [133]. It was found that the regioselectivity could be oriented to the “normal” optically active lactone **(−)-107** at a 90% yield with the mutant F255A/F443V, or to the “abnormal product” **(+)-195** in up to a 98% yield with the mutant variant W501V. The study also used **(+)-106** and **(−)-106** in the reactions to follow the regioselectivity of the reactions with different OTEMO mutants. The OTEMO wild type enzyme converts **(−)-106** to an equal (50:50) mixture of normal and abnormal lactones.

With cloned cyclohexanone monooxygenase into *Escherichia coli* TOP10[pQR239], Baeyer-Villiger catalyzed oxidation of racemic **106**, which gave an equimolar mixture of (−)-1(*S*),5(*R*)**-107** and its enantiomer (**−**)-1(*R*),5(*S*)**-107** on a pilot scale fermentation process [134,135], which were separated by low pressure chromatography (Scheme 44b). Further improvements in this reaction are expected to decrease the cost of the enantiomeric lactones and also that of the abnormal oxidation product.

### 2.4. The Use of δ-Lactones for Obtaining 9β-Halo-Prostaglandins and 9β-Halogenocyclopentane Intermediates

In Section 2.2, the halogen was used for its intramolecular substitution by the carboxylate anion with the formation of the γ-lactone ring. In this section, the halogen is kept unsubstituted in order to be used for obtaining 9β-halogenated prostaglandins or other halogenated cyclopentane intermediates. A lot of 9 (9α or 9β) or 11 (11α or 11β) halogenated prostaglandins have cytoprotective [135,136] or anti-trombotic activity [137,138,139], have promising results in the treatment of fertility problems [140], or in reducing the intraocular pressure [141], or can mimic the keto group to have great affinity to the PGE_2_ [137], PGD_2_ [136,137] receptors or the EP_2_ subtype receptor, which is the PGE_2_ receptor subtype [142]. These halogenated analogs were synthesized starting from the parent prostaglandin mainly by the following sequence: mesylation or tosylation of the hydroxyl group, SN_2_ substitution of the newly introduced good leaving group by the reaction with an inorganic or organic chloride, fluoride, or bromide salt [137,138,143,144,145,146]. Other alternatives were the Mitsunobu reaction [138,139] or fluorination with DAST [138] or diethyl (2-chloro-1,1,2-trifluoroethyl)amine [138] of the free hydroxyl of the specified prostaglandin.

A single example started from the chloro δ-lactone intermediate **196** (R = Bn, TBDMS, TBDPhS) for the synthesis of nocloprost, a 9β-chloro-prostaglandin analog, as depicted in Scheme 45. Let us notice that **196** has its chlorine atom in the correct position for obtaining 9β-chloro-prostaglandins, which is a position that was maintained during all steps of the nocloprost synthesis [143].

A different option was to use the δ-lactone intermediates **199**, having already the ω-side chain with the 15-keto group, to synthesize new 9β-halogenated prostaglandin analogs **200**–**201** by opening the lactone group with saturated diols (*n* = 2 to 5) or 2-butyne-1,4-diol in the presence of an acid catalyst, like TsOH, with or without an inert solvent, as presented in Scheme 46 [147,148].

These compounds have an ester group in position 6 of the prostaglandin α-side chain and an alcohol group instead of a C-1 carboxyl group, spaced by two to six methylene groups from the oxygen of the ester group. The best results were obtained for 1,4-butanediol. Notice that the α-side chain of this prostaglandin analogue mimics the length of the α-side chain of prostaglandins, but has a hydroxyl group instead of a carboxyl one [148]. The opening of the δ-lactone **202a**, with the allylic alcohol in the ω-side chain, proceeded at a very good yield (93% for **202a** to **203a**. For a similar reaction, ***epi*-202a** was slightly impurified with **203a**, but the total yield of ***epi-*203a** was also high) (Scheme 47). Compounds **202b** and **203b** (with 3-trifluorophnoxy instead of 3-chlorophenoxy) [148] were also synthesized.

The reduction of enone **200e** with diisobornyloxyaluminium isopropoxide, NaBH_4_-CeCl_3_ or NaBH_4_-Dowex 1 × 8 to the allylic alcohols **203a** and ***epi***-**203a** was not selective. The alcohols were formed in an almost 1:1 ratio. The separation of the isomers was performed by low pressure chromatography (LPC). The similar reduction of the δ-lactone intermediate **199a** proceeded in almost the same 1:1 ratio of allylic alcohols **202a** and ***epi*-202a**.

Compounds **200**–**203** were used in a molecular docking study to predict their potential cyto-protective (anti-ulcer) activity, with omeprazole (co-crystallized with the enzyme) and nocloprost being used as standards. All 9β-halogenated compounds and nocloprost with docking scores greater than that of omeprazole. The majority of the 9β-halogenated analogs (12 compounds) have a docking score greater than that of nocloprost, indicating that their cytoprotective (anti-ulcer) activity could be greater than that of nocloprost, which gave an impulse to further study the compounds [149].

A series of PGE_1_ analogues with an ester group in the 6th-position with mainly cytoprotective activity is also mentioned in the literature [150].

δ-Lactone intermediates of type **199** could be hydrogenated, as a step to obtain 13,14-hydrogenated prostaglandins. However, this hydrogenation is usually performed at the next step of the corresponding Corey compound with an ω-side chain. Hydrogenation of **199a** to **204a** was performed in >88.1% in an isolated compound after LPC purification (Scheme 48) [123].

δ-Lactones were also transformed into useful halogenated cyclopentane intermediates to be used as synthons in fine organic chemistry. Previously, δ-lactone **114** and the protected lactones with ether or ester groups like **116** were used to transform in the Corey γ-lactones with a different protection at the primary alcohols, especially with an ether group, followed by a sequence of reactions to protect the secondary alcohol and free the primary alcohol group or protect it with a complementary protecting group to be used in the next steps of the convergent prostaglandin synthesis to build the ω-side and α-side chain in the desired order. The only role of the halogen was to be intramolecularly substituted by the carboxylate anion, obtained by alkaline hydrolysis of the ester group, and to close the γ-lactone ring.

The intermediate halogenated cyclopentane compounds with four chiral carbon atoms are valuable to other chemical directions. Therefore, their synthesis, isolation, and protection of the secondary and primary alcohol groups is desirable.

The acid methanolysis of the benzoate lactone **116** gives mainly ester **117** with an amount of unprotected compound **115**, which increases with the reaction time. In a 1:1 mixture of methanol-CH_2_Cl_2_, the reaction goes cleanly to **117** at a quantitative yield in two days at rt. *p*-Nitrobenzoate give the corresponding compound **117a** at a 94.5% yield. Acetate **116b** is also partially hydrolyzed with **117b** being obtained at a 68.6% yield (see Scheme 49 and Table 4) [81,82]. The unprotected δ-lactone **114** was methanolized quantitatively to the chloro ester **115**.

Chloro esters **117** were protected at the secondary alcohol with an ether protecting group (TBDMS, trityl, THP) to **207**. After the deprotection of the ester group, the primary alcohol is protected with the same or with a different ether group to **208**. The primary alcohol group of chloro ester **115** was selectively protected to **205** with TBDMS and trityl or protected at both hydroxyl groups with the same protecting group. It is worth pointing out that the yields for obtaining different protected 5-chloro-cyclopentane compounds were high.

Other useful chloro ester compounds **211** [151] were obtained in the transesterification of the compounds **210** (obtained from δ-lactone **127** (Scheme 27) in two steps: **127**→**128**→**210** [92]), unprotected or then protected as TBDMS, benzoate, and THP, with K_2_CO_3_ in methanol (Scheme 50). The TBDMS and benzoate groups were lost during the reaction and chloro ester **211a** was obtained as an oil in a good yield (*n* = 0, 93.0% and 84.5% yield from **210a**, *n* = 1, 70.1% and 75.4% from **210b**). While the loss of the benzoate group was an expected outcome of the reaction, the loss of the TBDMS protective group was less anticipated. However, there are mentions in the literature about unexpected deprotection of TBDMS in base conditions [152,153]. The transesterification of bis-THP protected chloro esters **210c** gave the THP protected chloro esters **211c** in a good yield (*n* = 0, 83%, *n* = 1, 72%). The chloro esters **211a** could be protected at the secondary alcohol with the needed ether or ester group. The chloro esters **115**, **117**, **205**, **207,** and **211** could be used for building new compounds with these chloro-cyclopentane scaffolds.

## 3. Conclusions

The paper describes the synthesis of three types of lactones encountered in the total stereo-controlled synthesis of PGs: 1,9-, 1,11-, and 1,15-lactones, δ- and γ-Corey lactones. 1,15-PG-macrolactones of types of E_2_, E_3_, F_2_, F_3_, A_2_, and A_3_ were found in the marine molluscs *Tethys fimbria* and were synthesized to understand their role in the organism. Other 1,9-,1,11-, and 1,15-PG lactones of natural PGs and their analogs, especially drugs, were synthesized as an alternative to esters and also as intermediates for obtaining 5-trans free drugs. The methods for their synthesis are explicitly described in each case.

As key intermediates for the convergent synthesis of PGs by the Corey procedure, δ-lactones and γ-lactones represent a milestone for introducing the three or four stereo-centers on the cyclopentane fragment necessary for hydroxyls and for chemical groups to link to the PG side chains. We first lay the foundation and carefully outline the standard and most popular methods for the synthesis of lactones, starting from different racemic or optically active raw materials. We then present improvements of these standard methods, and new promising methods, interesting from the practical and/or from the research point of view. We analyze new interesting catalyzed enantio-selective or stereo-controlled reactions, chemical, or enzymatic resolution of the compounds in different steps of the sequences to PGs. Throughout the presentation, the common goal we follow is to obtain optically active lactones. The paper aims to give the reader a comprehensive overview of the lactones in the field of the prostaglandins, starting from the naturally occurring lactones, continuing with their synthesis (to gain a better understanding of the vital role they play in the metabolism, and to study their applications as a replacement for the C-1 esters and for the preparation of 5-trans-free prostaglandin isomers), and progressing to the key δ-lactone and γ-lactone intermediates in the milestone Corey strategy for obtaining all series of prostaglandins, in leukotrienes and isoprostane prostaglandin analogs. We also review the use of δ-lactones for obtaining halogen-substituted cyclopentane intermediates with interesting configurations, as synthons in the synthesis of 9β-halogeno-PG analogs (mainly chloro), and also for future applications.

## Data Availability

Not applicable.

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
