# Peer review of "Lactones in the Synthesis of Prostaglandins and Prostaglandin Analogs"

_ijms, 2021, doi:10.3390/ijms22041572_

Round 1

Reviewer 1 Report

In this manuscript, the authors aim to describe construction and preparation of lactones in the synthesis of prostaglandins and their analogues.  Since the syntheses of prostaglandins and their analogues have been discussed in the several reviews (Chem. Rev. 1993, 93, 1533; Chem. Rev. 2007, 107, 3286; Org. Biomol. Chem. 2017, 15, 6281), the authors intended to focus on the lactone synthesis in this manuscript.  However, not-a-few topics are overlapped with the previous reviews (For example, Scheme 8, 12, 30, 31, 32, 33, and 34).  The authors' focus is not concentrated, and the any lactone-related things and prostaglandins are unreasonably coupled and displayed in this manuscript.  In addition, the significance of each study was not well discussed.   Contributors (corresponding author) of the previous works are not clearly mentioned in this manuscript, which is inappropriate as a review.  The many typos and incorrect chemical structures (For example, olefins in Scheme 8) significantly lowered the readability.  According to the above points, I recommend rejection of the manuscript.  

Author Response

Dear Reviewer,

Thank you for your observations which we corrected, please see below:

At the observations: “Since the syntheses of prostaglandins and their analogues have been discussed in the several reviews (Chem. Rev. 1993, 93, 1533; Chem. Rev. 2007, 107, 3286; Org. Biomol. Chem. 2017, 15, 6281), the authors intended to focus on the lactone synthesis in this manuscript. However, not-a-few topics are overlapped with the previous reviews (For example, Scheme 8, 12, 30, 31, 32, 33, and 34).  The authors' focus is not concentrated, and the any lactone-related things and prostaglandins are unreasonably coupled and displayed in this manuscript.  In addition, the significance of each study was not well discussed.  Contributors (corresponding author) of the previous works are not clearly mentioned in this manuscript, which is inappropriate as a review. The many typos and incorrect chemical structures (For example, olefins in Scheme 8) significantly lowered the readability.”,

we respond as follows:

The Schemes mentioned above are presented in the manuscript in a modified form, not in that form presented in the referenced papers, to put in evidence those methods developed in high yields, from different raw materials, the applications of regio- and enantioselective reactions, the enzyme steps for obtaining optically active compounds, as well as some ingenious strategies for lactones, with potential value for future applications. 

- Scheme 8 was modified, according also to the observation of Reviewer 3 for the cis-correlation of the double bond in the α-side chain.

- Schemes 12 and 13 are present in the manuscript to evidence how a detailed study of step-by-step reaction metathesis resolved ingeniously the problem of obtaining the isoprostane intermediates with the α-side chain 66 and 69, otherwise not obtainable through lactone intermediate like 59 (Scheme 11).

- Scheme 30 is based on the original paper and is different from that presented in reference [83]

- Scheme 31 was presented in a modified form.

- Scheme 32 is slightly simplified.

- Scheme 33 was modified, taking into account also the observations of Reviewer 2.

- Scheme 34 was also presented due to the access to the all-cis-Corey lactone 175, used in the synthesis of isoprostanes, from the isomer tricyclic ketone 169 by a route with well-known high yield reactions.

The contribution of the corresponding author in previous works is mentioned in the manuscript, as follows:

- on pages 16-17 [references 87-88]: an efficient and selective transformation of δ-lactones 114 and 116 into the g-lactones 109 and 120.

- on page 18 [reference 92]: the transformation of δ-lactones 127 into aldehydes 129-130, protected as ketals or thioketals, crystallized and stable for years.

- on pages 28-29 [references 148-149]:

  • the transformation of δ-lactones 199, with w-side chain, into the 15-enone 9β-prostaglandin analogs 200-201 with an ester group at C-6 carbon atom and hydroxyl group at C-1.
  • the reduction of enone 199a and also of 15-enone 9β-prostaglandin analog 200e to the allylic alcohols 202a and epi-202a, respectively to 203a and epi-203a.
  • the transformation of δ-lactone 15-allylic alcohol prostaglandin intermediates 202a and epi-202a, by acid catalysis opening with diols, into the new 9β-prostaglandin analogs 203a and epi-203a.

- on page 29 [reference 125]: the selective hydrogenation of prostaglandin intermediate 199a to 204a.

- on page 29 [reference 150]: a molecular docking study of the 9β-halogenated prostaglandin analogs to predict their potential anti-ulcer (cytoprotective) activity was performed which revealed that 12 compounds had a docking score greater than that of nocloprost, used as standard; all compounds and nocloprost presented a docking score greater than that of omeprazole, used as standard in the study.

In conclusion, the contribution of the corresponding author is explained in the manuscript and we think this is a point worth mentioning in response to the reviewer’s comments.

The manuscript was corrected and the minor English mistakes and the typos were removed.

Reviewer 2 Report

Tanase and coworkers summarized the synthesis of lactones used in the synthesis prostaglandins and its analogues. As prostaglandins are important biologically active compounds and there are many synthetic methods for its preparation. Corey lactone is one of the well-known intermediates for the prostaglandins. Other lactones are also used as intermediates for the synthesis of the prostaglandins. There are several reviews of the synthesis of the prostaglandins. The present review focuses the lactones as a synthetic intermediate.

There are too many mistakes in English. The native speaker should check the English.

There are too many mistakes in the schemes.

10 in Figure 2: it should be mentioned the position of C17.

Scheme 1: it is very difficult to understand the position of proton.

Figure 3: 1,15-lactone of latanoprost: there is no double bond in the omega side chain in latanoprost.

Scheme 7: compound 35: An arrow is strange.

Scheme 8: compounds 42, 43:  What is the double bond in alfa-side chain?

Scheme 15: What is Hlg?  “X “is enough.

Scheme 30: from 142 to 143: What is red “1” of DMAP?

Scheme 33: check the structure of 161 and 162. The method of drawing the stereochemistry from 163 to 168 is different from the others. Please check it.

Scheme 35: from 177 to 178: 0.8 mol should be described as mol%.

Scheme 35: structure of 5-f3t-IP should be written.

Table 3:  It is difficult to understand this table. The table should be modified.

It is necessary to check all schemes carefully.

As there are so many mistakes, this reviewer does not support the publication of this review.

Author Response

Dear Reviewer,

Thank you for your valuable observations. The manuscript was corrected as follows:

- Observation 3: “10 in Figure 2: it should be mentioned the position of C17”. The position of C-17 was mentioned in the Figure.

- Observation 4: “Scheme 1: it is very difficult to understand the position of proton”. The formula of the intermediate was redrawn.

- Observation 5: “Figure 3: 1,15-lactone of latanoprost: there is no double bond in the omega side chain in latanoprost”. The formula of 1,15-lactone of latanoprost was introduced separately, without the double bond in the w-side chain.

- Observation 6: “Scheme 7: compound 35: An arrow is strange”. The mechanism was presented in the reference [39, page 114-116].

- Observation 7: “Scheme 8: compounds 42, 43:  What is the double bond in alfa-side chain?”. Scheme 8 was modified, as explained also in the comments for Reviewer 3.

- Observation 8: “Scheme 15: What is Hlg?  “X “is enough”. the caption of Scheme 15 was modified as follows: (I or Br at the C6 carbon atom in 74)

- Observation 9: “Scheme 30: from 142 to 143: What is red “1” of DMAP?.  The Scheme was modified; 1 is the first generation Grubbs’catalyst 145a; in both procedures a and b, 143 was obtained in the same yield.

- Observation 10: “Scheme 33: check the structure of 161 and 162. The method of drawing the stereochemistry from 163 to 168 is different from the others. Please check it”. Scheme 33 was corrected and modified a little bit in agreement with the observation of the reviewer.

- Observation 11: “Scheme 35: from 177 to 178: 0.8 mol should be described as mol%”. Indeed, it is mol %.

- Observation 12: “Scheme 35: structure of 5-f3t-IP should be written”. The formula of 5-F3t-IP was introduced in Scheme 35.

- Observation 13: “Table 3:  It is difficult to understand this table. The table should be modified”. Some explanations about the content of Table 3 were introduced.

- Observation 14: “It is necessary to check all schemes carefully.”. Thank you for your observations. We checked and corrected the Schemes, and corrected some mistakes.

- Observation 15: “As there are so many mistakes, this reviewer does not support the publication of this review.”. We hope that now, you would support the publication of the revised manuscript.

Reviewer 3 Report

This manuscript reviewed the synthesis of three different types of lactones in asymmetric synthesis of prostaglandins and prostaglandin analogs: 1,9-, 1,11- and 1,15-lactones, δ- and γ-Corey lactones. 

The authors summarized different methods for the synthesis of those lactones, such as Corey-Nicolaou method, Yamaguichi method, Shiina method, Mukaiyama method, and ring-closing metathesis. Expect for traditional lactonization method, the authors also included the use of enzymes for synthesis of optically pure lactone intermediates. 

This review paper is comprehensive and informative, I believe it should be published after minor revision:

  1. Page 1, line 35, "encountered" should be changed to "present"
  2. Page 2, line 54, "beenovserved" should be "been observed"
  3. page 2, line 54, "encountered" has been used too many times in this manuscript, other word choices should be considered, line 61, "encountered" should also be changed
  4. page 2, line 58, "inthe" should be "in the"
  5. page 2, line 60, "forthe" should be "for the"
  6. page 2, line 61, "subsequentapplications" should be " subsequent applications"
  7. Page 8, scheme 7, the arrows are huge, should be fixed. 
  8. Page 9, scheme 8. structure 42 and 43. the double bond after hydrogenation with lindlar catalyst should be cis. The geometry should be shown. 
  9. Scheme 18, compounds 96 and 07 are catalysts, should be drawn in a box or parenthesis to avoid confusion. 
  10. Scheme 29, 40, and 41, , all structures are too colorful. The style is not consistent with other schemes. Color should be used to highlight. 

Author Response

Dear Reviewer,

Thank you for your valuable observations. The manuscript was corrected as follows:

- Observations 2, 4, 5 and 6 (pause between the words) were corrected in the manuscript and not evidenced in a color.

- Observation 7: “Page 8, Scheme 7, the arrows are huge, should be fixed.”. The mechanism was presented in the reference [39, page 114-116].

- Observation 8: “Page 9, scheme 8. structure 42 and 43. the double bond after hydrogenation with Lindlar catalyst should be cis. The geometry should be shown”. The Scheme 8 was modified according to the suggestions of the reviewer.

- Observation 9: “Scheme 18, compounds 96 and 07 are catalysts, should be drawn in a box or parenthesis to avoid confusion.”. The catalyst 96 and 97 were presented in parenthesis. The catalysts were presented in parenthesis also in the other Schemes 11, 25, 30.

- Observation 10: “Scheme 29, 40, and 41, , all structures are too colorful. The style is not consistent with other schemes. Color should be used to highlight.”. The structures in Schemes 29, 40, 41 (and also in 38, 39, 42, 43) were modified and only the desired enantiomer was colored.

Round 2

Reviewer 1 Report

I am really concerned that the many points raised before are still not amended in the revised version.  Most importantly, the structure of the manuscript is completely the same to that of the previous version. At least, the totally different topics are discussed in the section 2.1 and sections 2.2-2.4 and unreasonably coupled as the one review.  These topics should be two separate reviews unless the corrections by better bridging.  Furthermore, there is not reply to the reviewers' comments in the cover letter.  This version is not acceptable unless the authors address the above points.

Author Response

Thank you for your observations.

In our previous Response, we tried to provide point-by-point answers to your observations (the answers were also included in the cover letter to the editor, but it seems that there must have been some error when introducing the answer in the on-line journal system). Please find them below.

Regarding the past observations: “Since the syntheses of prostaglandins and their analogues have been discussed in the several reviews (Chem. Rev. 1993, 93, 1533; Chem. Rev. 2007, 107, 3286; Org. Biomol. Chem. 2017, 15, 6281), the authors intended to focus on the lactone synthesis in this manuscript. However, not-a-few topics are overlapped with the previous reviews (For example, Scheme 8, 12, 30, 31, 32, 33, and 34).  The authors' focus is not concentrated, and the any lactone-related things and prostaglandins are unreasonably coupled and displayed in this manuscript.  In addition, the significance of each study was not well discussed.  Contributors (corresponding author) of the previous works are not clearly mentioned in this manuscript, which is inappropriate as a review. The many typos and incorrect chemical structures (For example, olefins in Scheme 8) significantly lowered the readability.”,

we responded as follows:

The Schemes mentioned above are presented in the manuscript in a modified form, not in the form presented in the referenced papers, to put in evidence those methods developed in high yields, from different raw materials, the applications of regio- and enantioselective reactions, the enzyme steps for obtaining optically active compounds, as well as some ingenious strategies for lactones, with potential value for future applications. 

- Scheme 8 was modified, according also to the observation of Reviewer 3 for the cis-correlation of the double bond in the α-side chain.

- Schemes 12 and 13 are present in the manuscript to evidence how a detailed study of step-by-step reaction metathesis resolved ingeniously the problem of obtaining the isoprostane intermediates with the α-side chain 66 and 69, otherwise not obtainable through lactone intermediate like 59 (Scheme 11).

- Scheme 30 is based on the original paper and is different from that presented in reference [83]

- Scheme 31 was presented in a modified form.

- Scheme 32 is slightly simplified.

- Scheme 33 was modified, taking into account also the observations of Reviewer 2.

- Scheme 34 was also presented due to the access to the all-cis-Corey lactone 175, used in the synthesis of isoprostanes, from the isomer tricyclic ketone 169 by a route with well-known high yield reactions.

The contribution of the corresponding author in previous works is mentioned in the manuscript, as follows:

- on pages 16-17 [references 87-88]: an efficient and selective transformation of δ-lactones 114 and 116 into the g-lactones 109 and 120.

- on page 18 [reference 92]: the transformation of δ-lactones 127 into aldehydes 129-130, protected as ketals or thioketals, crystallized and stable for years.

- on pages 28-29 [references 148-149]:

  • the transformation of δ-lactones 199, with w-side chain, into the 15-enone 9β-prostaglandin analogs 200-201 with an ester group at C-6 carbon atom and hydroxyl group at C-1.
  • the reduction of enone 199a and also of 15-enone 9β-prostaglandin analog 200e to the allylic alcohols 202a and epi-202a, respectively to 203a and epi-203a.
  • the transformation of δ-lactone 15-allylic alcohol prostaglandin intermediates 202a and epi-202a, by acid catalysis opening with diols, into the new 9β-prostaglandin analogs 203a and epi-203a.

- on page 29 [reference 125]: the selective hydrogenation of prostaglandin intermediate 199a to 204a.

- on page 29 [reference 150]: a molecular docking study of the 9β-halogenated prostaglandin analogs to predict their potential anti-ulcer (cytoprotective) activity was performed which revealed that 12 compounds had a docking score greater than that of nocloprost, used as standard; all compounds and nocloprost presented a docking score greater than that of omeprazole, used as standard in the study.

In conclusion, the contribution of the corresponding author is explained in the manuscript and we think this is a point worth mentioning in response to the reviewer’s remarks.

The manuscript was corrected and the minor English mistakes and the typos were removed.

Regarding the present observations:

“I am really concerned that the many points raised before are still not amended in the revised version.  Most importantly, the structure of the manuscript is completely the same to that of the previous version. At least, the totally different topics are discussed in the section 2.1 and sections 2.2-2.4 and unreasonably coupled as the one review.  These topics should be two separate reviews unless the corrections by better bridging.  Furthermore, there is not reply to the reviewers' comments in the cover letter.  This version is not acceptable unless the authors address the above points.”,

we respond as follows:

The paper aims to give the reader a comprehensive overview of the lactones in the field of the prostaglandins, starting from the naturally occurring lactones, continuing with their synthesis (to gain a better understanding of the vital role they play in the metabolism, and to study their applications as a replacement for the C-1 esters and for the preparation of 5-trans free prostaglandin isomers), progressing to the key δ- and g-lactone intermediates in the milestone Corey strategy for obtaining all series of prostaglandins, in leukotrienes and isoprostane prostaglandin analogs, and ending with the applications of δ-lactones for obtaining new 9β-halogeno-prostaglandins and halogeno-intermediates (for fine organic chemistry and future research).

This was the theme of the review from the very beginning, proposed to and accepted by the Guest Editors: Prof. Czeslaw Wawrzebczyc and Prof. Dr. Witold Gladkowski. We introduced a similar paragraph like the one above in the Introduction and also in the Conclusions of the manuscript.

We introduced also at Section 2 the following paragraph (red in the manuscript): In this paper we present the synthesis of the lactones encountered in prostaglandin and leukotriene synthesis. We include the synthesis of the milestone δ- and g-lactones used in the total stereocontrolled Corey procedure for the synthesis of prostaglandins and their analogs, and their isoprostane isomers. The chemical separation of pure enantiomers through diastereoisomer salts with optically pure amines or by enzymatical procedures, and also highly enantioselective catalyzed reactions for obtaining enantiopure compounds are presented in the paper. Finally, we treat also the synthesis of 9β-halo-prostaglandins and 9β-halo-cyclopentane intermediates from δ-lactones. These directions and finally Conclusions are presented in the plan outlined below:

The style arrangement of the captain 2, at the beginning and in the text, have been done.

Hence, we consider that there is a strong correlation between Section 2.1 and Sections 2.2-2.4, and it is in agreement with the main purpose and the plan of the manuscript.

Reviewer 2 Report

This reviewer thanks the authors to revise the manuscript according to the suggestions of this reviewer.

This reviewer would like to point out the following issues.

Scheme 33, compounds 163-168: The stereochemistry of the bridgehead should be written. What is the relative stereochemistry in 161b and 162?

In Table 3, the followings issues should be considered.

What is R4 = Tr + 3-OTr?

What is R4 =TBDMS + 3-OTHP?

Why is [alpha] D necessary for this table?

The melting point is “oil”. This is strange.  Is the description of oil necessary?

Author Response

Answer to Reviewer 2.

Thank you for your valuable observations. We corrected the manuscript in agreement with your observations, as follows:

  1. Scheme 33, compounds 163-168: The stereochemistry of the bridgehead should be written. What is the relative stereochemistry in 161b and 162? The stereochemistry of the compounds 163-168 was presented in the reference [106a] with points to the carbon atoms in the bridge. In the ESI of the reference [106a] the stereochemistry is the one which was modified in Scheme 33.

Regarding the stereochemistry of alcohols 161b and 162: the alcohols are a mixture of diastereoisomers, and they are presented so in the Scheme. See also pages 22-23 for minor additions.

  1. In Table 3, the followings issues should be considered.

What is R4 = Tr + 3-OTr?

What is R4 =TBDMS + 3-OTHP?

Why is [alpha] D necessary for this table?

The melting point is “oil”. This is strange.  Is the description of oil necessary?

In the renumbered Table 4, in fact it is Tr + 5-OTr, for the bis-tritylated compound 205c,

                                                     TBDMS + 5-OTr, for compound 205e.

 -The column with mp was removed; sure, it was a mistake. Thank you for your observation!

- [alpha] D was introduced to make these data easily available in this review, otherwise they would be more difficult to find in the patent.

This idea was suggested at the end of 2020 by a reviewer for another manuscript who could not find the experimental data for the starting compound in the pending patent, so he requested us to introduce the same experimental procedure in the manuscript.

Indeed, even with SciFinder many times it is difficult to find analytical data, like [alpha] D, so I think that including [alpha] D in the Table is useful for the readers.

Thank you once more for your observations and for carefully reading our manuscript.

Round 3

Reviewer 1 Report

I understood the thought of authors and editors.

At least, the incorrect format (for example, curved allows in Scheme 7) and incorrect spelling (for example, Furtner) must be amended.

Author Response

Dear Reviewer,

Thank you for your valuable observations.

We corrected the manuscript in agreement with your observations:

-the incorrect spelling for Furtner, line 260 in Fürstner

-in Fig. 7 (page 9 line 245) we removed a curved arrows between H-O to O

Yours sincerely

Constantin Tanase